

# Cadmium nitrate and DNA methylation in gastropods: comparison between ovotestis and hepatopancreas

George A. Draghici[1,2], Cristina A. Dehelean[1,2], Alina E. Moaca[1,2], Marius L. Moise[3], Iulia Pinzaru[1,2], Valentin N. Vladut[4], Ioan Banatean-Dunea[5] and Dragos Nica[2,4]

[1] Department of Toxicology and Drug Industry, Faculty of Pharmacy, University of Medicine and Pharmacy of Timisoara, Timisoara, Timis, Romania

[2] Research Center for Pharmaco-Toxicological Evaluations, Faculty of Pharmacy, University of Medicine and Pharmacy of Timisoara, Timisoara, Timis, Romania

[3] Premiere Hospital, Regina Maria Health Network, Timisoara, Timis, Romania

[4] The National Institute of Research –Development for Machines and Installations Designed for Agriculture and Food Industry, Bucharest, Romania

[5] Faculty of Agriculture, University of Life Sciences "King Mihai I" from Timişoara, Timisoara, Timis, Romania

Corresponding authors
Cristina A. Dehelean,
cadehelean@umft.ro
Dragos Nica, nicadragos@gmail.com

## ABSTRACT

Dietary ingestion is the main route of exposure to hazardous contaminants in land animals. Cadmium, a high-profile toxic metal, affects living systems at different organismal levels, including major storage organs (liver, kidneys), key organs for species survival (gonads), and epigenetic networks regulating gene expression. 5-methylcytosine (5mC) is the most common and best-characterized epigenetic mark among different modified nucleosides in DNA. This important player in methylation-driven gene expression is impacted by cadmium in sentinel terrestrial vertebrates. However, limited information exists regarding its impact on macroinvertebrates, especially land snails commonly used as (eco)toxicological models. We first investigate the methylomic effects of dietary cadmium given as cadmium nitrate on terrestrial mollusks. Mature specimens of the common brown garden snail, *Cornu aspersum*, were continuously exposed for four weeks to environmentally-relevant cadmium levels. We determined global genomic DNA methylation in hepatopancreas and ovotestis, as well as changes in the methylation status of CG pairs at the 5′ region close to the transcription site of gene encoding the Cd-selective metallothionein (Cd-MT). Weight gain/loss, hypometabolism tendency, and survival rates were also assessed. Although this exposure event did not adversely affect survival, gastropods exposed to the highest Cd dose revealed a significant reduction in body weight and a significant increase in hypometabolic behavior. The hepatopancreas, but not the ovotestis, displayed significant hypermethylation, but only for the aforementioned specimens. We also found that the 5′ end of the *Cd-MT* gene was unmethylated in both organs and its methylation status was insensitive to cadmium exposure. Our results are important since they provide scientists, for the first time, with quantitative data on DNA methylation in gastropod ovotestis and refine our understanding of Cd epigenetic effects on terrestrial mollusks.

## INTRODUCTION

Environmental contaminants affect epigenetic mechanisms regulating gene expression, including DNA modifications, histone variants and their modifications, and non-coding ribonucleic acid (RNA) molecules (*Srut, 2021*). Besides the different post-translational modifications (PTMs) in histone proteins (*e.g.*, methylation, phosphorylation, acetylation), DNA methylation of the genome (methylome) is the most common form of post-replicative epigenetic modifications in eukaryotes (*Head, Dolinoy & Basu, 2012*) and involves the attachment of methyl (-CH$_3$) groups to DNA (*Parry, Rulands & Reik, 2021*). This biological process occurs mainly at cytosine-phosphate-guanine (CpG) sites, especially at the 5′-carbon of cytosine residues—5-methylcytosine (5mC). This DNA methylation variant is the best-characterized DNA modification in animals and plays a key role for long-term, stable gene repression (*Head, Dolinoy & Basu, 2012*; *Luo, Hajkova & Ecker, 2018*). Changes in genome-wide 5mC levels have been used as biomarkers in vertebrate sentinel organisms to monitor the adverse effects of different environmental contaminants, including highly toxic trace metals such as cadmium (Cd), lead (Pb), or mercury (Hg) (*Cavalieri, 2020*). The epigenetic impact of anthropogenic pollution on terrestrial macroinvertebrates is, however, still not well understood (*Radwan et al., 2019*). The brown garden snail, *Cornu aspersum* (Muller, 1774), used as a model organism, exhibits remarkable tolerance to long-term, high-dose exposure to cadmium (Cd) and other toxic trace metals (*e.g.*, lead, mercury) without showing major metabolic side effects (*Dallinger et al., 2001*).

Cadmium is a persistent pollutant that is easily transferred along terrestrial trophic chains and accumulates to substantial amounts in both invertebrates and vertebrates (*Nica et al., 2017a*). The liver (hepatopancreas in mollusks and arthropods) and kidneys serve as the main storage sites. However, other organs, such as muscles, bones, and gonads, can also retain this metal at levels above environmental concentrations (*Dallinger et al., 2001*; *Rankin, 2009*). For example, terrestrial pulmonate gastropods *Bradybaena fruticum* (Muller, 1774), *Aegopis verticillus* (Lamarck, 1822), *Arianta arbustorum* (Linnaeus, 1758), and *Helix pomatia* (Linnaeus, 1758) collected from an area with long history of pollution in Arnoldstein (Austria) showed median hepatopancreas cadmium concentrations of 300 milligrams per kilogram dry weight basis (mg/kg d. wt Cd) for soil levels ranging between 8.9 mg/kg d. wt Cd and 26.6 mg/kg d. wt Cd. The measured concentrations in kidneys and foot muscles reached values up to 100 mg/kg d. wt Cd and 30 mg/kg d. wt Cd, respectively (*Rabitsch, 1996*). At natural levels of soil Cd (0.48−1.24 mg/kg d. wt), mature specimens of *Helix vladika* (Kobelt, 1898) and *Helix secernenda* (Rossmassler, 1847) obtained from three sampling sites in Montenegro displayed cadmium concentrations up to three-fold higher in the reproductive system relative to the soil (*Vukasinovic-Pesic et al., 2020*). Similar data were reported for other ecologically-relevant gastropod species, including *C. aspersum* (*e.g.*, *Menta & Parisi, 2001*; *Carbone & Faggio, 2019*).

Toxic effects of Cd on land snails include growth restriction, hypometabolic responses, reproductive side-effects, and organ toxicity (*Dallinger et al., 2001*). At the molecular level, this metal can cause oxidative stress, DNA damage, and epigenetic changes. Available evidence indicates that dietary exposure as either cadmium chloride (CdCl$_2$) or
cadmium sulfate ($CdSO_4$) is associated with changes in global DNA methylation levels in *C. aspersum* hepatopancreas (*Nica et al., 2017a*; *Georgescu et al., 2021*). In contrast, there is no information about the impact of other cadmium compounds on gastropod methylome. A high-profile example is cadmium nitrate (($CdNO_3$)$_2$). This potent carcinogen is commonly used in manufacturing of cadmium salts, laboratory reagents, photographic emulsions, and colored glass (*Lu et al., 2015*). Similar to the aforementioned salts, it is highly soluble in water (*Rankin, 2009*), thereby facilitating Cd biomagnification along terrestrial food chains. However, to the best of our knowledge, its epigenetic effects have yet to be investigated in animal phyla.

Literature data show that cadmium does accumulate in the hermaphrodite glands (*syn.* ovotestis) of gastropods (*Vukasinovic-Pesic et al., 2020*; *Dhiman & Pant, 2021*). This organ is of utmost importance for species survival (*Dallinger et al., 2001*) and possible transgenerational effects related to exposure to environmental contaminants (*Osborne, Gillis & Prosser, 2020*; *Wang & Liu, 2023*). A major drawback for toxicological studies aiming to assess the methylomic effects of environmental contaminants on ovotestis is the lack of data about the 5mC content of this key organ.

Until recently, invertebrate genomes were thought to display, as a general rule, mosaic methylation, with regions of heavily methylated DNA interspersed with non-methylated regions; and gene-body methylation (GBM), with methylation marks found primarily in the coding regions (exons) and non-coding regions (introns) of the gene transcriptional part (gene body) (*Manner et al., 2021*). In contrast, available evidence indicates that vertebrate genomes are predominantly heavily methylated (*Angeloni & Bogdanovic, 2021*) and display cytosine DNA methylation at CpG dinucleotides, except for CpG sites from genomic regions with high CpG ratios and GC contents—regions also known as "CpG islands" (CGIs) (*Al Adhami et al., 2022*). However, the paradigm of DNA methylation in invertebrates has been recently challenged as growing evidence shows that it is more sophisticated than previously assumed (*De Mendoza et al., 2019*; *Klughammer et al., 2022*). A recent comparative analysis of DNA methylation profiles across 580 animal species (including 45 invertebrates) revealed that the total amount of 5mC in the genome of many invertebrates species is well within the levels seen in vertebrates (*Klughammer et al., 2022*). The haplosclerid demosponge *Amphimedon queenslandica* (Hooper & Van Soest, 2006), for example, showed a hypermethylated genome, with methylation levels and patterns showing strong similarities with vertebrates (*De Mendoza et al., 2019*). Moreover, certain invertebrates (*e.g.*, sea urchins) display the DNA methylation profiles specific to vertebrates; that is, low gene promoter methylation and elevated gene-body methylation (*Klughammer et al., 2022*).

CpG islands frequently lie at the 5′-end of genes and contain DNA sequences to which relevant proteins (*e.g.*, RNA polymerase, transcription factors) bind to begin transcription (promoters); and sites where the first DNA nucleotide is translated into RNA (transcription start sites) (*Al Adhami et al., 2022*). The methylation status/patterns of these gene-regulatory elements can regulate gene expression in vertebrates, especially in environment-responsive genes (inducible genes), and can be affected by Cd exposure (*Arita & Costa, 2009*; *Sanders et al., 2013*). A number of studies have identified CGI-like features

and described CpG-dense regions surrounding the 5′-end of genes in invertebrate clades (*Angeloni & Bogdanovic, 2021*). Pertinent data suggests that DNA methylation patterns across these genomic regions could be important for epigenetic regulation of molluskan genome, *e.g.*, in the Pacific oyster, *Crassotrea gigas* (Thunberg, 1793), or in *Aplysia* sea slugs (*Rajasethupathy et al., 2009*; *Riviere, 2019*). There is also indication that CG pairs at the 5′ region close to the transcription site of gene encoding the Cd-selective metallothionein (*Cd-MT*) in *C. aspersum* hepatopancreas are constitutively unmethylated and not affected by Cd exposure (*Georgescu et al., 2021*) although this gene is inducible and pivotal for $Cd^{2+}$ detoxification and stress response (*Hispard et al., 2008*; *Hockner et al., 2011*; *Palacios et al., 2011*). Given the critical importance of ovotestis for gastropod survival, resistance, and persistence (*Dallinger et al., 2001*; *Osborne, Gillis & Prosser, 2020*; *Wang & Liu, 2023*), we also thought that is important to find whether cadmium affects the methylation status of these CG pairs in the ovotestis of *C. aspersum*. There is evidence, although rare, that the CG pairs at the 5′-end of invertebrate genes can be methylated and responsive to environmental factors, but this effect might be limited to selected genes and tissues. Such an example in gastropods is a CpG island in the promoter of the *CREB2* gene from *Aplysia* neurons (*Rajasethupathy et al., 2009*).

Our main objective was to determine the effect of dietary cadmium given as cadmium nitrate on selected DNA methylation parameters in *C. aspersum*, that is global 5mC levels and the methylation status of CG pairs at the 5′ region of the *Cd-MT* gene in the hepatopancreas and ovotestis of adult brown garden snail, *Cornu aspersum*. Elucidating the effect of cadmium nitrate on selected DNA methylation parameters in land snails will aid our understanding of how mollusks are able to cope at the epigenetic level with environmental pollution. The results of the present study will also enlarge the present knowledge of invertebrate methylome by providing scientists with the first data on global DNA methylation in gastropod ovotestis. Using epigenetic approaches in an (eco)toxicological context is an emerging trend with broad implications for environmental toxicology with respect to experimental methodologies and chemical risk assessment (*Srut, 2021*).

## MATERIALS & METHODS

### Experimental design and rearing conditions

All experiments were run at the Laboratory Animal Facility from the ''Victor Babes'' University of Medicine and Pharmacy, Timisoara. In September 2021, 160 newly-matured specimens of the common brown garden snail, *Cornu aspersum*, were purchased from the ''Mokry Dwor'' snail farm (Krzymow wielkopolskie, Poland). Throughout the four-week experimental period, the gastropods were fed *ad libitum* with Cd-spiked, agar-based diets. *Ad libitum* feeding implies that experimental animals had these diets available to them all the times; hence were fed without restriction. This type of feeding management is the most often used method for different animal models in dietary exposure experiments for chemicals in food (*World Health Organization, 2009*).

The fodder was prepared as previously described (*Nica et al., 2017a*). That is, the control diet was prepared by mixing 50 grams (g) infant cereals (Nestle Nestum 5—Five

Cereals), 20 g carrot baby food (HiPP, UK), and 15 g agar (A-1296, Sigma) with double distilled water to obtain 1000 milliliters (mL) agar medium. After adding three milliliters of 1% methylparaben solution to extend the storage period, each liter of medium was divided equally among forty 7-cm-diameter Petri dishes (about 25 mL/dish). These plates were maintained post-cooling in the refrigerator for maximum one week. In the case of Cd-spiked diets, the fodder was prepared in a similar manner, with solutions of different cadmium concentrations being used instead of double distilled water. The eight nominal Cd treatments used to prepare the fodder, that is 0, 0.05, 0.1, 0.2, 1, 5, 10, and 100 milligrams per liter (mg/L), were abbreviated as Cd0, Cd0.05, Cd0.1, Cd0.2, Cd1, Cd5, Cd10, and Cd100. Cadmium nitrate tetrahydrate ($Cd(NO_3)_2$ $4H_2O$, $\geq$ 99.99% trace metal basis, Sigma-Aldrich) was used as a source of cadmium.

Exposures were conducted in two independent experiments per each treatment group (Replica I and Replica II). Each replica consisted of 10 specimens. For each replicate jar, the snails were maintained in polypropylene containers ($50 \times 20 \times 20$ cm) with perforated side walls at 18 to 20 °C, under a constant light-dark cycle of 12:12 h. The bottom of each container was covered with ash-free filter paper sheets ($50 \times 20$ cm), which were wetted twice daily with double distilled water. Food was given daily (two Petri dishes per container). The daily activity schedule involved fodder supply, removal of faeces and uneaten food, replacing the sheet of ash-free filter paper, cleaning the containers with sterile paper towels, and, of particular importance, snail fitness monitoring and dead specimen collection.

Before running experiments, the shell height was measured with a digital caliper to the nearest 0.01 mm (for all snails) as the distance between the suture and the base of expanded lip (*Kerney & Cameron, 1979*). The most homogeneous four gastropods from each replicate were selected based on this parameter; that is, eight specimens per treatment group ($n = 8$). These specimens were weighed with an analytical balance to the nearest 0.01 mg in order to provide us with the same level of uniformity among sampling groups in terms of pre-exposure body weight. Their shells were numbered with black acrylic paint to ensure proper identification of each gastropod.

At the end of the experiment, the numbered snails were weighed again to determine the post-exposure body weight. For each experimental group, four gastropods (two specimens per each replicate) were randomly sacrificed to collect hepatopancreas and ovotestis samples for DNA extraction. Results from previous studies lend support to this sample size ($n = 4$) as being enough for detecting changes in total 5mC content of gastropod DNA (*Muller et al., 2016*; *Sanchez-Arguello et al., 2016*; *Bal, Kumar & Nugegoda, 2017*; *Nica et al., 2017a*; *Georgescu et al., 2021*). In addition, such sample sizes are not unusual for exploratory studies aiming to determine trends or patterns of changes (*Ko & Lim, 2021*), which was the case of our investigation. Moreover, this moderate sample size is often used in (eco)toxicological studies on genome-wide DNA methylation due to budget and time constraints.

## Snail fitness

Snail fitness was evaluated based on sublethal and lethal toxicological endpoints. Weight gain/loss and hypometabolism tendency were used as sublethal toxicological endpoints.

The former variable was determined as the difference between the snail weight at the beginning and the end of the experiment. The latter variable was defined as the tendency of snails to exhibit reduced metabolic activity (stop/reduce feeding and enter into aestivation). Survival rate was used as a lethal endpoint of exposure.

## DNA extraction and 5mC quantification analysis

Genomic DNA was extracted from hepatopancreas and ovotestis with the GenElute™ Mammalian Genomic DNA Miniprep Kit (Merck KGaA, Darmstadt, Germany) as per manufacturer's instructions. A DS-11 Spectrophotometer (DeNovix, Wilmington, USA) was used to evaluate DNA purity by determining a sample absorbance spectrum at 260 nm and 280 nm, and calculating the $A_{260}/A_{280}$ ratio. Genome-wide 5mC levels in DNA samples were assessed using the MethylFlash Global DNA Methylation (5-mC) ELISA Easy Kit (EpiGentek, East Farmingdale, NY, USA), as previously described (*Georgescu et al., 2021*). Optical density (OD) was measured at 450 nanometers using a Stat Fax 4200 microplate reader (Awareness Technology, Palm City, FL, USA). For absolute 5-mC quantification, a standard curve was obtained by plotting the various concentrations of the positive controls against the corresponding ODs.

## Methyl specific polymerase chain reaction (MSP)

The effect of cadmium (given as cadmium nitrate) on the methylation status of the seven CpG and GpC sites in the 5′-UTR promoter region of the *Cd-MT* gene in the hepatopancreas of mature *C. aspersum* specimens was investigated. MSP analysis was run using one forward primer (CdMTbs-F1/ggatttatYGtaggatattaattaagg) and two reverse primers (CdMTbs-R1/ctaaaaataaaaccaaataccaatcctac; CdMTbs-R2/Ccttaccacacttacaaccatc) (*Georgescu et al., 2021*). The bisulfite-converted DNA was obtained after using the 'high molarity' protocol (*Genereux et al., 2008*). More precisely, 20–25 microliters of DNA were first denatured in 0.3 N sodium hydroxide (NaOH) at 42 °C for 20 min (min). At the same time, bisulfite solution was prepared under a hood by dissolving 2.10 g sodium bisulfite (NaHSO$_3$) and 0.68 g ammonium sulfite monohydrate (($NH_4)_2SO_3$ $H_2O$) in five mL of 45% $NH_4HSO_3$ in water (all chemicals from Merck, Darmstadt, Germany) to obtain a final volume of 6.0 mL and a pH of 5.4. Denatured DNA samples were then put into 185 microliters bisulfite solution and maintained at 70 °C for 150 min.

The bisulfite-converted DNA samples and the HotStarTaq Master Mix reagent (Qiagen, Hilden, Germany) were subjected to PCR amplification with the aforementioned primers. For each PCR, 0.25−0.5 micrograms of DNA were used; the reaction volume was 40 microliters. PCR conditions were: denaturation for 15 min at 95 °C; four cycles of denaturing for 30 s (sec) at 94 °C, annealing for 35 s at 54 °C, and elongation for 30 s at 72 °C; then 5 min at 72 °C. To verify for specific amplification by MSP, a 10 microliters aliquot of the PCR product was run on a 1.5% Tris-acetate-EDTA-agarose gel (Merck, Darmstadt, Germany). After agarose gel electrophoresis, the amplicons were visualized by using ethidium bromide staining (Merck, Darmstadt, Germany).

## Statistical analysis

First, the homogeneity of different treatment groups with respect to pre-exposure body weight was anayzed. These data sets were verified for normality and homoskedasticity (homogeneity of variance) using Anderson-Darling tests and Bartlett's tests, respectively. A one-way ANOVA was applied if both these assumptions were met, with post hoc comparisons using the Scheffe's procedure being applied against the control group in case of significant results. If the data did not meet the aforementioned criteria, a Kruskal-Wallis test with Dunn's post tests was used.

Next, the effect of dietary cadmium on absolute values for weight gain/loss was determined using a similar approach. After that, the snails were classified into specimens with and without hypometabolic behavior. These data were tabulated as a $8 \times 2$ contingency table and analyzed using a $Chi^2$ goodness of fit test, with tests based on $2 \times 2$ contingency tables being run against controls in case of significant differences. Finally, snail survival across different treatments was compared using a Log-Rank test. Post hoc comparisons using the Breslow's procedure were applied *versus* controls in case of significant results.

The proportion of snails surviving at the end of the experimental period for each treatment was determined using a Kaplan–Meier curve. The mean survival time was calculated as the area under the Kaplan–Meier estimate of the survival curve. The treatment was considered complete for the dead gastropods and censored for the specimens who were still alive at the end of the study.

The genome-wide methylation data sets were transformed by decimal logarithmation and then analyzed in the same manner as the weight-related data. A T test on methylation data for controls was next applied to determine whether there are significant differences between the total 5mC contents in the DNA samples from ovotestis and hepatopancreas. For bisulfite PCR, statistical analysis was performed separately for each of the seven CpGs studied. After being classified into methylated and unmethylated, these data were analyzed in a similar way to data sets for hypometabolism tendency. All statistical analyses were conducted with the Statistica 8 software package (StatSoft Inc., Tulsa, OK, USA) and a two-tailed $p$ value <0.05 was considered significant.

# RESULTS

## Snail fitness

Mean pre-exposure body weight, the number of snails with hypometabolic behavior, and the mortality-related data for each treatment group are given in Table 1. Data sets for the former variable met the criteria for normality (Anderson-Darling tests, $p \geq 0.327$) and homogeneity (Bartlett's test, $p = 0.962$). The average weighted pre-exposure body weight for all groups was 8.89 g. The highest values were seen in controls and the lowest values in the Cd10 snails (Table 1). The raw data used to generate these values are given in Table S1. There were no significant inter-group differences in pre-exposure body weight (ANOVA, $(F_{(7, 56)} = 1.12, p = 0.361)$). These results validate our experimental assumption, *i.e.*, the gastropods selected for this study were of uniform body weight.

Weight gain/loss at the end of the experiment ranged from $-0.43$ g in the Cd100 gastropods up to 2.64 g in the Cd0.1 specimens, as seen in Fig. 1. The overall weighted

**Table 1 Pre-exposure body weight, hypermetabolism tendency, and mortality for each treatment group.** Data for pre-exposure body weight (second column to the left) are given in grams (g) as mean value with one standard deviation (in parenthesis). These data were measured for eight specimens per each treatment group. Data on hypometabolic behavior (third and fourth columns to the left) are shown as the number of snails (nr.) and the corresponding percentage (%). Data related to mortality are presented as the number of dead snails (second column to the right) and the corresponding mortality percentages (first column to the right).

| Tretament | Pre-exposure weight (g) | Hypometabolism tendency | | Mortalities | |
|---|---|---|---|---|---|
| | | (nr.) | % | (nr.) | % |
| Cd0 | 8.32 (0.71) | – | 0% | – | 0% |
| CD0.05 | 8.57 (0.94) | – | 0% | – | 0% |
| Cd0.1 | 8.91 (0.81) | – | 0% | – | 0% |
| Cd0.2 | 8.72 (0.85) | – | 0% | – | 0% |
| Cd1 | 9.28 (1.01) | – | 0% | – | 0% |
| Cd5 | 8.97 (1.11) | 1 | 5% | – | 0% |
| Cd10 | 9.23 (0.81) | 3 | 15% | 1 | 5% |
| Cd100 | 9.12 (0.89) | 10 | 50% | 2 | 10% |

mean for all groups was 1.80 g. The raw data used to create this figure are displayed in the Table S1. These data sets were normally distributed (Anderson-Darling tests, $p \geq 0.209$) and homoskedastic (Bartlett's test, $p = 0.122$).

Cadmium exerted a significant impact on weight gain (ANOVA, ($F(7, 56) = 21.74$, $p < 0.001$)). Specimens from the lowest four treatment groups showed similar weight gains compared to controls (Fig. 1). There was a trend of decrease in body weight gain starting from the third highest dose onward (Fig. 1). However, significant differences occurred only for gastropods receiving the highest cadmium dose (Cd100); that is, a significant 5% decrease in body weight (Fig. 1).

Three weeks after starting the experiment, a part of gastropods from the three highest treatment groups tended to reduce/cease feeding and/or enter into aestivation as cadmium dose increased (see Table 1). The proportion of snails with hypometabolism tendency differed significantly between groups (Chi² test, $df = 7$, $p < 0.001$). Post hoc comparisons against controls yielded significant results for the specimens receiving the highest Cd dose (Chi² test, $df = 1$, $p < 0.001$), but not for the other treatment groups (Chi² tests, $df = 1$, $p \geq 0.073$).

No mortalities occurred in controls, as well as for the numbered snails. Low death rates were found for cadmium-exposed specimens (see Table 1). In addition, no significant inter-group differences were observed in temporal distribution of death after cadmium exposure (Log-Rank test, $p = 0.927$). The raw data used to conduct this analysis are given in the Table S2.

## Genome-wide DNA methylation levels

Log$_{10}$-transformed data for total 5mC levels in DNA of the hepatopancreas and ovotestis are shown in Figs. 2 and 3, respectively. The corresponding absolute values are given in Table S3. The mean values in hepatopancreas ranged from 0.27% in specimens exposed
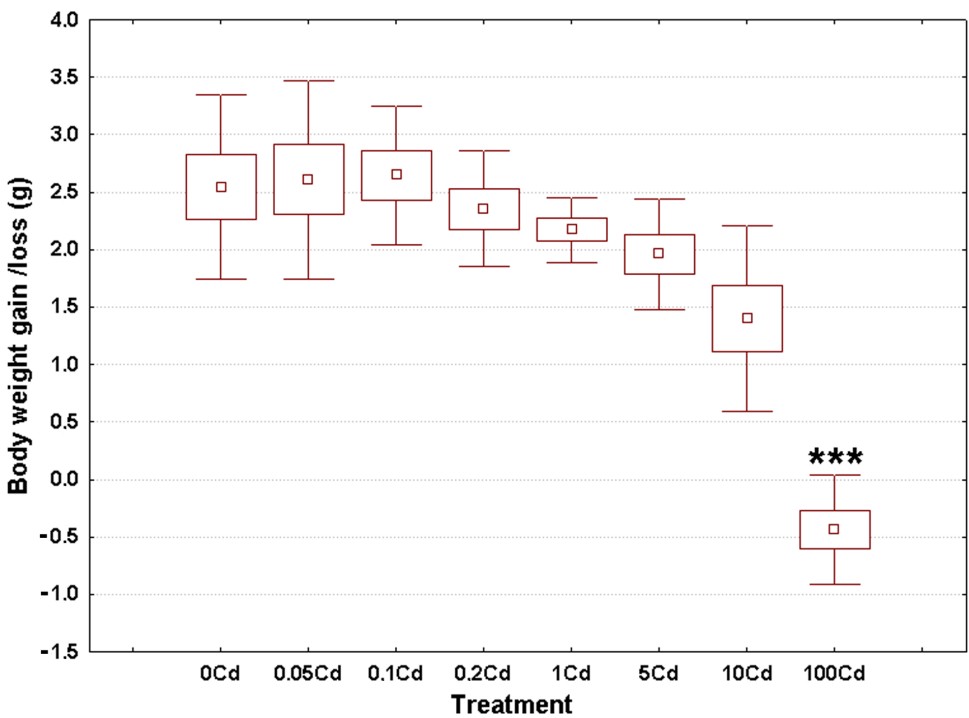

**Figure 1  Effect of dietary cadmium on body weight changes ($n = 8$).** Data sets are shown as mean (point) with one standard error (box) and one standard deviation (error bar) (Scheffe's tests, ***—$p < 0.001$, **—$p < 0.01$, *—$p < 0.05$).

to the second lowest Cd dose (Cd0.1) up to 0.88% in snails fed the highest cadmium dose (Cd100), with an overall weighted mean for all groups of 0.38% (Fig. 2). The average ovotestis 5mC content varied between 0.42% in gastropods from the second lowest treatment group (Cd0.1) and 0.94% in specimens from the highest treatment group (Cd100). The corresponding overall weighted mean for all groups was 0.62% (Fig. 3).

All data sets met the assumptions of normality (Anderson-Darling tests, $p \geq 0.273$) and homogeneity of variance (Bartlett's, $p \geq 129$). Significant changes occurred in genomic DNA methylation levels in snail hepatopancreas (ANOVA, ($F(7, 31) = 4.47$, $p = 0.002$)), but not in snail ovotestis (ANOVA, ($F(7, 31) = 0.13$, $p = 0.162$)). Post hoc testing revealed a significant hypermethylation of hepatopancreas DNA for snails exposed to the highest Cd dose (Cd100). In contrast no effect was seen for the other treatments (Fig. 2).

There was no evident dose–response pattern with respect to the 5mC content of ovotestis DNA (Fig. 3). However, the total DNA methylation levels in the ovotestis of controls were significantly greater than the values measured in the hepatopancreas (T test, $p = 0.023$). This trend was also observed in Cd-exposed snails (Figs. 2 and 3). The mean variance (for pooled methylation data sets) in ovotestis was almost three-fold higher than that seen for the hepatopancreas (0.13 *vs* 0.5).

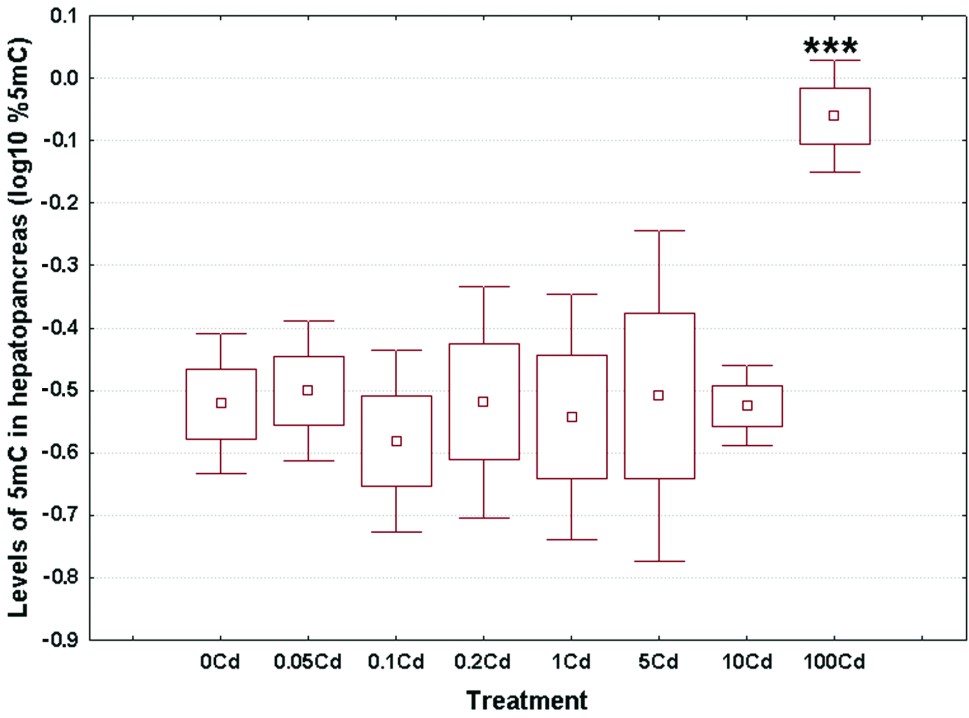

**Figure 2  Effect of dietary cadmium on genome-wide DNA methylation levels in hepatopancreas ($n = 4$).** Methylation data are shown on a $\log_{10}$ scale. Data sets are depicted as mean (point) with one standard error (box) and one standard deviation (error bar) (Scheffe's tests, ***—$p < 0.001$, **—$p < 0.01$, *—$p < 0.05$).

### Cd-MT gene promoter methylation

For all treatment groups, all cytosines from the seven analyzed CG base pairs were converted to thymines after bisulfite treatment and PCR amplification, irrespective of the primer pairs used and organ analyzed. Cadmium dose had no significant effect on their methylation status (Chi$^2$ test, $df = 7$, $p = 1$). These data suggest that the 5′ region close to the presumed *Cd-MT* transcription start site is not likely to be constitutively methylated in the hepatopancreatic DNA and ovotestis DNA, and its methylation status is not affected by cadmium exposure.

## DISCUSSION

The contribution of this investigation is threefold: first, it significantly expands available knowledge of DNA methylation in gastropods by determining, for the first time, the ovotestis 5mC content. Second, this is the first study to analyze the impact of cadmium given as cadmium nitrate on animal methylome, thus filling a gap in environmental epigenetics. Third, it refines current understanding about the practical applicability of terrestrial mollusks as sentinel organisms for monitoring ecotoxicological effects of environmental pollution by comparing the relevance of various endpoints at different levels of biological organization.

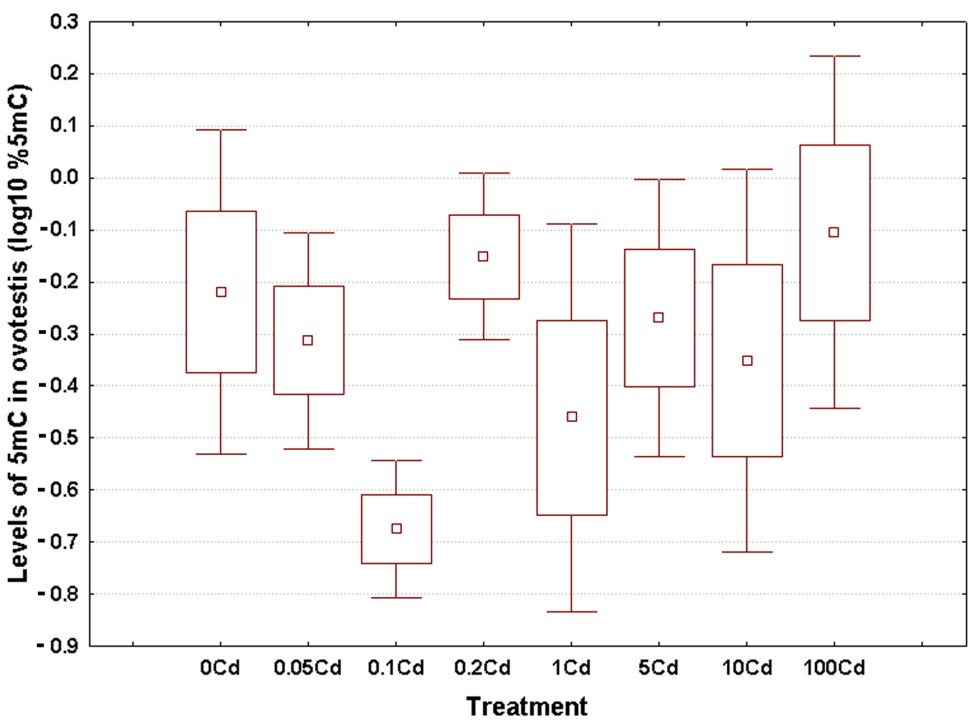

**Figure 3  Effect of dietary cadmium on genome-wide DNA methylation levels in ovotestis ($n = 4$).** Methylation data are shown on a $\log_{10}$ scale. Data sets are depicted as mean (point) with one standard error (box) and one standard deviation (error bar) (Scheffe's tests, ***—$p < 0.001$, **—$p < 0.01$, *—$p < 0.05$).

## Snail fitness

Gastropod exposure to cadmium yields adverse effects on multiple fitness endpoints, such as body weight, shell growth, and metabolic activity (*Russell, De Haven & Botts, 1981*; *Laskowski & Hopkin, 1996*; *Gomot, 1997*; *Nica et al., 2015*; *Nica et al., 2017b*). Mature *C. aspersum* snails were fed diets containing a broad range of Cd concentrations, including the maximum allowable levels in vegetal foods routinely consumed by land snails, such as leafy vegetables or fruits (*MWFEP, 2002*; *European Comission* No 1881/*2006*); and the reference values for hepatopancreatic cytotoxicity and genotoxicity (*Hodl et al., 2010*). The present study hence provides a realistic perspective of the potential (eco)toxicological implications related to exposure to cadmium nitrate.

Under the present experimental conditions, cadmium given at these environmentally relevant concentrations exerted a significant effect on body weight only in mature snails exposed to very high doses (at least 100 mg/kg d. wt Cd). However, the pattern observed in the dose–response curve indicates that smaller exposure doses may affect body weight for longer exposure durations. It is also possible that the sample size used in this study ($n = 8$, for body weight) was not large enough to detect statistically significant differences.

Interestingly, the same specimens displayed a significant increase in hypometabolism tendency. However, this exposure event did not adversely affect survival. These findings are in agreement with literature data showing that decrease in body weight, feeding cessation,

and entrance to dormancy precedes lethality in Cd-exposed gastropods (*Russell, De Haven & Botts, 1981*; *Laskowski & Hopkin, 1996*; *Gomot, 1997*; *Nica et al., 2015*). Moreover, this lack of major effects on fitness parameters critical for survival reinforces the utility of this land snail species as invertebrate model for dissecting the pathobiology of sublethal cadmium exposure.

## Cadmium effect on gastropod DNA methylation

Generally speaking, genomic DNA methylation levels in the hepatopancreas and ovotestis of *C. aspersum* are comparable to those quantified *via* ELISA-based procedures in the whole body, foot, or head of the southern creeper, *Zeacumantus subcarinatus* (Sowerby, 1855); the acute bladder snail, *Physa acuta* (Draparnaud, 1805); and the ram's horn snail, *Biomphalaria glabrata* (Say, 1818) (*Joe, 2013*; *Muller et al., 2016*; *Geyer et al., 2017*; *Aparicio, 2021*). The measured values in the hepatopancreas of controls are also consistent with those determined in *C. aspersum* specimens of similar age and size (*Nica et al., 2017a*; *Georgescu et al., 2021*). Taken together, these findings suggest that the global 5mC content in gastropod DNA is likely to exhibit moderate inter-species and inter-organ variation, that is 0.1−4.9% as determined *via* ELISA (*Joe, 2013*; *Muller et al., 2016*; *Geyer et al., 2017*; *Nica et al., 2017a*; *Aparicio, 2021*; *Georgescu et al., 2021*); and lie at the lower end of the range reported for invertebrates (*Srut, 2021*).

Congruent with literature data on cadmium sulphate (*Georgescu et al., 2021*), only specimens given 100 mg/kg d. wt Cd as cadmium nitrate displayed significant hypermethylation of hepatopancreas DNA. In the case of cadmium chloride, this response occurred from doses of 10 mg/kg d. wt Cd onward (*Nica et al., 2017a*). Different values of acid dissociation constants of these salts may account for these outcomes since there is indication that different anions bound to the same cation within a salt exert distinct DNA methylation changes (*Gao et al., 2013*).

Global 5mC content of ovotestis DNA was, by contrast, not affected by cadmium uptake *via* food. This organ retains smaller amounts of cadmium when compared to the hepatopancreas (*Vaufleury & Kerhoas, 2000*; *Nica et al., 2017a*; *Vukasinovic-Pesic et al., 2020*), and thus may exhibit a lower Cd sensitivity. It is also plausible that the higher variability of total 5mC percentage in the former organ is related to the different stages of snail reproductive cycle. For example, gametogenesis is associated with dynamic and continually changing genome-wide DNA-methylation levels required to generate mature sperm and ovocytes (*Maamar et al., 2022*). With respect to this organ, we also note that the measured values in controls were significantly above those in the hepatopancreas. However, the biological significance of all these findings cannot be precisely established at this stage without confirming these results and additional experimental evidence.

Cytosine methylation in hepatopancreas DNA was absent at the 5′-end of the *Cd-MT* gene and MSP experiment did not allow us to observe changes in the CpG methylation level in response to oral Cd given as cadmium nitrate. This is in line with previous findings relating to land snails exposed to cadmium sulfate (*Georgescu et al., 2021*). Similar results were obtained for the ovotestis. Analysis of DNA methylation status in CG pairs of the Cd-inducible metallothionein gene *wMT-2* promoter in the common earthworm, *Lumbricus*

*terrestris* (Linnaeus, 1758), also revealed that these sites are normally unmethylated. This was also valid for Cd-exposed specimens (*Drechsel et al., 2017*). As a result, one can expect a low likelihood of 5mC to occur at the promoter region of the *Cd-MT* gene in *C. aspersum*. In fact, literature data indicate that functional DNA methylation at gene promoters is a rare paradigm among mollusks. To the best of our knowledge, such a mechanism has been reported in gastropods only for the *Aplysia* sea slugs. That is, a CpG island in the promoter of the *CREB2* gene—a key gene in long-term memory formation and persistence—commonly exists in both unmethylated and methylated states and its methylation status is responsive to serotonine exposure (*Rajasethupathy et al., 2012*). There is also evidence, although again rare, for functional promoter DNA methylation in bivalve mollusks; *e.g.*, methylation of CpG4-6 in promoter region of the gene encoding galectin in the pearl oyster *Pinctada fucata* (Gould, 1850) (*Li et al., 2015*).

### Relevance of DNA methylation parameters in land snails as toxicological endpoints

To date, toxicological knowledge of cadmium nitrate impact on animals was limited to data derived from enzymatic, cytotoxic, genotoxic, and organismal endpoints (*e.g.*, *Ledda-Columbano et al., 1984*; *Lin et al., 1994*; *Yamadori et al., 2010*). The present findings provide a new perspective on its effects by showing that cadmium nitrate can interfere with epigenetic mechanisms regulating gene expression. This effect is accompanied by changes in other relevant exposure endpoints, such as body weight gain or hypometabolic tendency. Such responses are known to occur in terrestrial gastropods exposed to cadmium, but have never been reported for dietary cadmium given as cadmium nitrate.

Consistent with previous laboratory investigations (*Nica et al., 2017a*; *Georgescu et al., 2021*), the total 5mC content in DNA of *C. aspersum* hepatopancreas was sensitive to chronic continuous Cd exposure only for high dietary doses, which are rarely encountered under field conditions. Its sensitivity as a biomarker of sublethal stress was, however, similar to those observed for hypometabolism tendency and body weight changes. In addition, the other methylation parameters investigated here seemed to be unresponsive to oral cadmium given as cadmium nitrate. Taken together, these findings considerably disfavor the use of genome-wide DNA methylation levels in gastropods as biomarkers of cadmium exposure at levels relevant for environmental and human health. However, these data can provide us with a rough estimate of the lowest-observable-effect concentration (LOEC) of cadmium required for inducing changes in total 5mC content of the hepatopancreas; under the present experimental conditions, between 80 and 100 mg/kg d. wt.

The present experimental design did not allow us to determine whether the changes observed for the aforementioned variables share a common physiological background, derive one from another, or are totally independent. Although there is no information linking DNA methylation with metabolism in gastropods, there is relevant evidence for such an association in other invertebrates undergoing aestivation to endure adverse environmental conditions. Thus, the Japanese sea cucumber, *Apostichopus japonicus* (Selenka, 1867), exhibits intestinal hypometabolism during aestivation *via* DNA hypermethylation-mediated transcriptional suppression of metabolic pathways (*Li et*

*al., 2018*). Since the snails were maintained under controlled laboratory conditions (optimal for their well-being), it is plausible that this tendency towards reduced metabolic activity (stop/reduce feeding and enter into aestivation) is not related to desiccation or lack of fodder. Rather, this phenomenon may reflect Cd-induced changes in key enzymes of the DNA methylation machinery (*e.g.*, DNA methyltransferases, Ten Eleven Translocation enzymes, Methyl-CpG-binding domain proteins). Indeed, chronic exposure to cadmium can yield such effects by increasing the levels of DNA methyltransferases (DNMTs), as already described in rat cells (*Takiguchi et al., 2003*), and further lead to DNA hypermethylation, hypometabolism, aestivation, body weight reduction, and death. Moreover, recent data suggests that *C. aspersum* possesses functionally active *DNMT1* gene, but not *DNMT3* gene, and this gene is sensitive to cadmium exposure (*Draghici et al., 2020*).

The present study focused on cadmium, but did not determine the levels of other metals in food or snail body. Such measurements could be important since humans and wildlife are exposed under real-life conditions to complex mixtures of metals and chemicals, with exposure to a single metal/chemical rarely yielding a dominant effect. There is relevant evidence that cadmium feeding can perturb metal homeostasis in invertebrates and co-exposure to metal mixtures may interfere with the toxic effects of cadmium, potentially yielding synergistic effects. Thus, for cladocerans *Daphnia magna* (Straus, 1820) and *Moina macrocopa* (Straus, 1820), mixtures of copper (Cu), lead (Pb), Cd, nickel (Ni), and zinc (Zn) exerted stronger multi-generational effects on growth, reproduction, and population dynamics than single exposure to Cd (*Sun et al., 2021*). Moreover, gastropods fed Cd-spiked diets displayed altered Cu and manganese (Mn) balance (*Nica et al., 2019*).

These findings highlight the practical difficulties which arise when investigating the effect of cadmium and other environmental toxicants on DNA methylation in terrestrial snails. Moreover, Cd concentrations in food and vegetables of importance for animal and human nutrition are typically well below the levels causing significant changes in total 5mC content of gastropod DNA (*Panel, 2009*). However, one cannot exclude the presence of targeted changes in DNA methylation at specific genes, gene regions (*e.g.*, at gene bodies), and genomic areas in the ovotestis, hepatopancreas, or other organs for environmentally realistic levels of exposure. In fact, DNA methylation in the marine mollusk *Crassostrea gigas* (Thunberg, 1793) is responsive to environmental stress, especially in gene networks with inducible expression (*Gavery & Roberts, 2010*). In addition, specimens of freshwater snails *P. acuta* exposed to the fungicide vinclozolin (VZ) revealed consistent changes in regional DNA methylation patterns (*Sanchez-Arguello et al., 2016*). Unfortunately, it is currently not possible to conduct high-resolution DNA methylation analysis in land snails due to the limited number of reference genomes available, scarce information about gastropod methylome, and the presence of a partially sequenced transcriptome for the brown garden snail, *Cornu aspersum* (*Parmakelis et al., 2017*; *Fallet et al., 2020*). In this context, obtaining a complete genome sequence and annotation for *C. aspersum* and land snail species of (eco)toxicological importance, as well as sequencing and deciphering their methylomes, are sine qua non conditions to elucidate the potential of DNA methylation in

these mollusks to serve as toxicological endpoints for cadmium and other environmental contaminants.

Most studies to date have used ELISA (*Joe, 2013*; *Muller et al., 2016*; *Geyer et al., 2017*; *Nica et al., 2017a*; *Aparicio, 2021*; *Georgescu et al., 2021*), and much rarely liquid chromatography-tandem mass spectrometry (LC-MS/MS) and other chromatography-based methods (*Fneich et al., 2013*), for quantitative measurement of genome-wide DNA methylation in land snails. The later approach is regarded as the gold standard in methylation analysis, but is expensive and requires MS expertise despite being sensitive, quantitative, and reproducible (*Liu, Hesson & Ward, 2013*). On the other hand, ELISA-based methods are cost-effective and fast, but sensitive to the quality of DNA and prone to higher variability compared to chromatography- based methods (*Liu, Hesson & Ward, 2013*). As a result, future studies should use chromatography-based methods to validate ELISA results. Another possibility is to use the dot blot assay. This method has been already validated on other mollusk species, and as an advantage, it is more cost effective than ELISA-based methods (*Luviano et al., 2021*).

## CONCLUSIONS

Overall, the results from our laboratory study suggest that genomic DNA methylation in the hepatopancreas of mature *C. aspersum* snails is sensitive to oral cadmium given as cadmium nitrate, but only for very high exposure doses. A similar sensitivity to cadmium contamination is observed for body weight loss and hypometabolism tendency although such an exposure event does not adversely impact survival. Total DNA methylation in the ovotestis is less responsive to cadmium contamination. These findings provide scientists, for the first time, with pertinent information about the methylomic effects of cadmium nitrate on animal genomes; and quantitative data on DNA methylation in gastropod ovotestis, thus refining our understanding of Cd epigenetic effects.

### Funding
The authors received no funding for this work.

### Competing Interests
The authors declare there are no competing interests.

### Author Contributions
- George A. Draghici conceived and designed the experiments, performed the experiments, prepared figures and/or tables, authored or reviewed drafts of the article, and approved the final draft.
- Cristina A. Dehelean analyzed the data, authored or reviewed drafts of the article, and approved the final draft.
- Alina E. Moaca performed the experiments, prepared figures and/or tables, and approved the final draft.

- Marius L. Moise conceived and designed the experiments, analyzed the data, authored or reviewed drafts of the article, and approved the final draft.
- Iulia Pinzaru performed the experiments, prepared figures and/or tables, and approved the final draft.
- Valentin N. Vladut analyzed the data, prepared figures and/or tables, and approved the final draft.
- Ioan Banatean-Dunea conceived and designed the experiments, performed the experiments, prepared figures and/or tables, and approved the final draft.
- Dragos Nica conceived and designed the experiments, analyzed the data, authored or reviewed drafts of the article, and approved the final draft.

## Data Availability

The raw measurements are available in the Supplemental File.

## Supplemental Information

Supplemental information for this article can be found online at http://dx.doi.org/10.7717/peerj.15032#supplemental-information.

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
