# Peer review of "Cadmium nitrate and DNA methylation in gastropods: comparison between ovotestis and hepatopancreas"

_PeerJ, doi:10.7717/peerj.15032_

## Round 0.1 · original submission · Major Revisions

The reviewers have suggested many changes before the next round of review. Please incorporate all and re-submit.

·

Basic reporting

Graghici et al. reports here the impact of cadmium exposure for global/local DNA methylation level in hepatopancreas and ovotestis and for some relevant traits (Weight gain/loss, hypometabolism tendency, and survival rates) in the gastropod Cornu aspersum. The article is clear, well written and the english is professional.
The introduction is well detailed, sufficient field background is provided and the research questions are well defined. I suggest that the authors improve the description at lines 94- 97 to provide more recent relevant information about DNA methylation patterns in invertebrates. Indeed, the paradigm of an invertebrate DNA methylation pattern being only mosaic and specific to gene body has been recently shaken and several interesting articles relevant to the field may be worth to mention, this would underline the fact that DNA methylation in invertebrates is more sophisticated that previously anticipated.
De mendoza et al. Nat Ecol Evol 2019 doi: 10.1038/s41559-019-0983-2
Klughammer et al. bioRxiv 2022 doi.org/10.1101/2022.06.18.496602

Other information which could be relevant is the amount of Cadmium that is found in the environment or in storage sites. The appropriate literature is included in the introduction but the content of the articles could be mentionned with more details.
Exemple L78-80: The liver and kidneys serve as the main storage sites, but other organs, such as muscles, bones, and gonads, can retain this metal at levels (please give details) high above environmental concentrations (pleave give details).

A last point that could be relevant to mention in the introduction, is the choice of ovotestis for DNA methylation studies. I suggest that the authors include another critical importance for the ovotestis choice which is the possible transgenerational effect of the cadmium exposure which implicates this organs. (L116-118 Given the critical importance of ovotestis for species survival (Dallinger et al., 2001), we also thought that is important to find whether cadmium affects the methylation status of these CG pairs in the ovotestis of C. aspersum).

Experimental design

The methods are generally well described with sufficient details & information to replicate. I suggest that the authors improve the description at lines 135-136. What do the author mean by fed ad libidum ? Was the agar-based diet renewed every days? every other days? once every week? What quantity of food? Do the mollusks feed the same way ? Or are there variation in the amount of food ingested by the different individuals ? My question may be naive but I wonder if each mollusks were exposed at the same quantity of pollutants and this could explain some interindividual variability. I would appreciate some explanation on this part of the protocol.
Further explanation on the PCR condition will also be appreciated in the section describing the Methyl Specific Polymerase Chain Reaction.

Validity of the findings

The results are generally relevant to the field, worth to be published and the investigation have been conducted rigorously and to a high technical standard. I suggest that the authors pay more attention to the clear dose response effect of the cadmium exposure for the body weight gain/loss which is represented on the figure 1. Only the high dose of cadmium seems to lead to a statistically different result compared to control, but obviously a dose response effect is observed. Maybe the moderate sample size (4 samples) is at the origin of the absence of statistically significant differences?

The analysis on the DNA methylation studies performed with the ELISA kits are well conducted, more details of the experiment are given in the author’s previous paper (https://doi.org/10.3390/toxics9110306) and maybe this paper should be given as a reference for more details. The interindividual variability in ovotestis is well discussed in the discussion section and I totally agree with the authors on this point (L351-354).

I would be more cautious with the interpretation of the MSP experiment performed in the promoter of the Cd-MT gene. This MSP experiment is appropriate as a quick and reliable way to validate change in methylation in CpG sites as long as there is a positive control to validate this change. In this work, the authors don’t observe any changes in CpG methylation in response to Cadmium nitrate exposure but without any positive control (as in their previous paper adressing the effect of the Cadmium Sulfate). I am not sure that the experiment is resolutive enough to lead to such a conclusion : « Cytosine methylation in hepatopancreas DNA was absent at the 5’-end of the Cd-MT gene and this status did not change in response to oral Cd given as cadmium nitrate ». (L358-359). A more appropriate experiment would be to do bisulfite conversion of DNA, subsequent PCR and sequencing of the PCR product. However, this is worth that this kind of negative result appear in the paper since it supports the current literature as the authors have discussed (L365-366) but their conclusion should be dampen, maybe as follow « Cytosine methylation in hepatopancreas DNA was absent at the 5’-end of the Cd-MT gene and MSP experiment did not allow to observe changes in the CpG methylation level in response to oral Cd given as cadmium nitrate ». And the sentence (L440-441) should not be mentionned in the main conclusion as the result clearly do not support such a conclusion.

Additional comments

Please, for evidence of functional DNA methylation in promoter sequences of mollusk, I suggest to mention another work performed in oyster (doi: 10.1016/j.fsi.2015.06.016).
In the discussion about the relevance of DNA methylation parameters in land snail as toxicological endpoints, the possible synergistic effect with other pollutants may be worth to discuss. (doi: 10.1016/j.watres.2021.117274)
Last point, that the authors may consider in their disucssion (L429-432) is another method to study global DNA methylation : the dot blot assay as validated on other mollusk species, much more cost effective than ELISA based methods (doi: 10.1186/s13072-021-00422-7).

Reviewer 2 ·

Basic reporting

.

Experimental design

.

Validity of the findings

.

Additional comments

Major Comments
 Authors should clearly and briefly state which new toxic effect of cadmium nitrate
 The manuscript should be presented concisely in a scientific style in the sections “Introduction, Materials and Methods, Results, and Discussion”, suggestions to delete unnecessary text
 There are many self-plagiarism with Ref. Georgescu et al. (2021).
 Write factually and neutrally in the third person, not in the 1st person “we” or “I”.
Examples:
1. Page 7 Line 71: We used here………………………..
2. Page 11 L 204: First, we anayzed the homogeneity………
3. Page 11 L 210: Next, we determined…………..
4. Page 11 L 211: After that, we classified………………….
5. Page 11 L 214: Finally, we compared between ……………..
6. Page 12 L 256: we found that …………………
7. Page 13 L 290: we found that …………………
8. Page 15 L 392: Since we maintained………………….
Minnor Comments
Abstract
Page 7 L 49-50: Remove “, but only for the aforementioned specimens.”
Introduction
 Try to keep the introduction as short, concise, and simple as possible.
 Page 7 L 60: “, DNA” Changed to “. DNA”
 Page 7 L 71: (Radwan et al., 2018) corrected to (Radwan et al., 2019)
 Page 7 L 71-75: reconstruction of part “We used here the brown garden snail, Cornu aspersum (Muller 1774), as model organism. This snail …………………….” to “The brown garden snail Cornu aspersum, (Muller 1774), used as a model organism, and showing remarkable tolerance to long-term, high-dose exposure to cadmium (Cd) and other toxic trace metals (e.g., lead, mercury) without showing major metabolic side effects (Dallinger et al., 2001)”
 Page 7 L 78: “but” changed to “also”.
Materials and Methods
 Page 10 L 188- 195: It must be mention only the methods of the present study not the previous what the authors did. As the authors wrote in part “Methyl Specific Polymerase Chain Reaction (MSP)”. “Previous research has investigated ……………………….. sites.”
Results
In the results section, the current data should be briefly and concisely addressed.
Discussion
 Page 13 L312: “Regulation EC No 1881/2006” corrected to “European Comission No 1881/2006”.
 Page 14 L 354: Mention the reference details instead [32].
 Page 15 L 384: 100 mg/Kg d. wt. changed to “100 mg/Kg b.wt”.
 Page 16 L 412: Add reference “Gavery & Roberts, 2013” in the references list, and remove reference ”Gavery MR, Roberts SB. 2010. DNA methylation ………..” from the reference list or mention the right one.
References
 Page 17 L 445 - 605: Change the title of the Journals in most references in “References list” to full title not abbreviation to follow the journal's instructions.
 Page 18 L 499: Ref. “Head JA. 2012. Dolinoy, D.C.; Basu, N. Epigenetics for……………” Corrected to “Head JA, Dolinoy DC, Basu N. 2012…..”.
 There are some references cited in references section, and not cited in the text For Example:
1. Bal N, Kumar A, Nugegoda D. 2017.
2. Hispard F, Schuler D, de Vaufleury A, Scheifler R, Badot PM, Dallinger R. 2008.
3. Maamar MB, Beck D, Nilsson E, McCarrey JR, Skinner MK. 2022.
4. Nica DV, Draghici GA, Andrica FM, Popescu S, Coricovac DE, Dehelean CA, Gergen II,
5. Kovatsi L, Coleman, M.D.; Tsatsakis, A. 2019.
6. Parry A., Rulands S, Reik W. 2021.
7. Pierron F, Baillon L, Sow M, Gotreau S, Gonzalez P. 2014.

Reviewer 3 ·

Basic reporting

The abstract is well-written.
The introduction section needs to discuss the literature gap and net contribution of the study. Moreover, the discussions in the introduction section can have more details to highlight the importance of the research.

Experimental design

The methodology section needs to explain more about sample of the study. Overall, the methodology is well-described.

Validity of the findings

The validity of the findings is OK. However, the discussions of the study have room for improvement.

Additional comments

The overall quality of the research is good.

---

## Round 0.2 · accepted · Accept

The reviewers have accepted the paper. and I agree. Thus, the paper is accepted for publication.

·

Basic reporting

I have no further comments on the content of this manuscripts. The authors replies to all the comments that i had mentionned and i think the manuscript meets the peeJ criteria for acceptation

Experimental design

nothing else

Validity of the findings

nothing else